# Practical Adaptation of a Low-Cost Voltage Transducer with an Open Feedback Loop for Precise Measurement of Distorted Voltages

**DOI:** 10.3390/s19051071

**Published:** 2019-03-02

**Authors:** Maciej Sulowicz, Krzysztof Ludwinek, Jaroslaw Tulicki, Wojciech Depczynski, Lukasz Nowakowski

**Affiliations:** 1Faculty of Electrical and Computer Engineering, Cracow University of Technology, Warszawska 24 Street, 31-155 Cracow, Poland; jtulicki@pk.edu.pl; 2Faculty of Electrical Engineering, Automatic Control and Computer Science, Kielce University of Technology, Al. Tysiaclecia P. P. 7, 25-314 Kielce, Poland; k.ludwinek@tu.kielce.pl; 3Faculty of Mechatronics and Mechanical Engineering, Kielce University of Technology, Al. Tysiaclecia P.P. 7, 25-314 Kielce, Poland; wdep@tu.kielce.pl (W.D.); lukasn@tu.kielce.pl (L.N.)

**Keywords:** alternating current (AC) and direct current (DC) voltage measuring, Hall-effect sensor, voltage transducer, Finite Element Method Magnetics (FEMM), ferrite core, air gap, leakage inductance

## Abstract

This paper presents the project proposal of a low-cost transducer with a Hall-effect sensor placed in a ferromagnetic core’s air gap, which enables the measurement of the distorted voltage instantaneous values without the feedback loop used for measurements in electrical machines. The presented transducer allows for electrical separation between the measured voltage and the voltage at the output. Moreover, the influences of frequency, additional resistance, and the reactance of the winding circuit on the voltage phase shift caused by winding inductance with ferrite core and amplitude are discussed. The result of simulating leakage inductance of measuring winding with ferrite core with an air gap is calculated using finite element analysis. Experimental investigations of the voltage phase shift angle and output voltage amplitude drop for the voltage transducers with an open feedback loop are carried out, taking into account the linear core magnetization characteristic.

## 1. Introduction

In industry, voltage measurements are usually performed using electrical devices or optical sensors, using contact or contactless measurement methods [1,2,3,4,5,6,7,8,9,10,11,12,13,14]. Each of these methods has its advantages and disadvantages depending on the measured voltage type (alternating current (AC), or direct current (DC), or AC and DC), voltage amplitude and slope, frequency, and temperature. For high voltages, the most common methods are inductive voltage transformers, capacitive voltage transformers (CVTs or CCVTs), capacitive dividers, or optical sensors [1,3,7,8,9,10,11,12,13,14]. For low voltages, the most common methods are inductive voltage transformers, resistive dividers, optical voltage sensors, and Hall sensors [2,4,5,7,8,10]. Introducing a feedback loop in transducers significantly improves the accuracy of voltage measurement [12]. The measuring method with the use of optical voltage sensors is dynamically developing. Optical voltage sensors have lighter weight and advantageous features, such as a wider bandwidth (up to 10 GHz), larger electric field dynamic range (up to 1 MV/m), and resistance to electromagnetic interference [10,11,12,13,14]. Optical sensors can be costly; additionally, they require complex systems/circuits and, thus, they are seldom used in industry voltage measurement systems.

The applications of various voltage measurement methods are studied in many papers. At the low-frequency range, the example methods are non-intrusive AC line-to-neutral voltage measurements using stray capacitance in a one-phase isolated cable with three wires for 60 Hz [9], and with one wire in the range from 100 to 1000 Hz [7] and from 50 to 800 Hz [8]. In References [7] and [9], there were no results regarding the influence of the frequency of measured voltage on the phase-shift angle variation and the method of correction. The advantage of this method (using stray capacitance) is its non-intrusive measurement. However, the disadvantage is the influence of the wire placement with respect to the isolation position and of the wire cross-section on the accuracy of the voltage measurement.

The frequency influence of measured voltage signals on their phase-shift angle and amplitude variation is investigated in many papers [7,15]. However, for the authors, there are no known applications of amplitude-phase characteristics (frequency response) of Hall-sensor-based voltage measurement transducers for a correction of error caused by the frequency.

The application of integrated-circuit sensors placed in the air gap of a magnetic core for the measurement of AC and DC voltages, currents, and magnetic flux density in electrical machines is constantly being developed [1,2,16,17,18,19,20,21,22,23,24,25,26,27,28,29]. For AC and DC currents with a magnetic flux density, contact and contactless measurement, Hall-effect sensors, magnetoresistive current sensors, and coils (e.g., the Rogowski coils are only suitable for AC currents) [16,17,22,23,24] are commonly used. In cases of voltages up to 1 kV, with output separation, both magnetoresistive current sensors and Hall-effect sensors can be used after placing them in the air gap of a ferromagnetic core, and voltage measurement is carried out by using the current to measure magnetic flux in the measuring winding that is wound around the core [25,26,27]. The main disadvantages of magnetoresistive current sensors are their voltage offset, lack of electrical model, surrounding noise sources, temperature drift, and frequency bandwidth [1,26,27]. Apart from the Hall-effect and magnetoresistive current sensors (contact measurement), voltage measurement can be carried out by measuring the electric flux density [3]. This method is based on measuring the changes in capacitance (or electric charge) due to changes in electric flux density, and is applied only to high voltages [3]. In cases of voltages up to 1 kV, AC and DC voltage measurement contact probes and sensors dominate, e.g., a resistive voltage divider without output separation, and voltage transducers with open and closed feedback loops [1,2]. Simultaneous measurement of phase-to-neutral and phase-to-phase voltages using, for example, oscilloscopes or data acquisition (DAQ) cards, requires electrical circuit separation. Measurement of DC and AC voltages in electrical machines is most commonly done with voltage transducers with Hall-effect sensors working with a closed or an open feedback loop [4,18]. Voltage transducers with both closed and open feedback loops are available on the market [4]. In order to amplify the measured voltage, current, or magnetic field, the Hall-effect sensor is placed in the core air gap [2,7,8,22,23,24,25,26,27,28]. The magnetic flux density shape in the core and in the air gap depends on the electric and magnetic properties of the core and on the range of measured frequencies. Due to hysteresis, the relation between the magnetic flux density *B* and the magnetic field strength *H* is considered using complex magnetic permeability [30,31].(1)$$\underline{\mu }=\mu e^{-j\psi }=\mu_{r}-j\mu_{i},$$
where *ψ* is the phase-shift angle resulting from over-magnetization losses, and *μ_r_* and *μ_i_* are the real and imaginary parts of the complex magnetic permeability, respectively.

In modelling, the occurrence of the real and imaginary parts of magnetic permeability is the cause of the phase shift between the magnetic flux density *B* and the magnetic field strength *H*. The presence of the imaginary part of magnetic permeability *μ* appears only above several dozen kHz, causing an increase in the value of the magnetic permeability in the initial course of the characteristic *μ* = *f*(*H*) [32,33]. In the low-frequency range, up to several dozen kHz, the imaginary part of magnetic permeability is negligibly small [32,33]. The presence of an alternating electromagnetic field in the core is manifested by the occurrence of the so-called skin effect [32,33,34]. The skin effect is related to the penetration depth of the electromagnetic field *λ* in the core. The higher the penetration depth is, the greater the electrical conductivity γ, the magnetic permeability *μ_c_* of the core, and the frequency *f* are. The depth of field penetration is expressed formulas follows:(2)$$\lambda =\frac{1}{\sqrt{\pi f\gamma \mu_{c}\mu_{0}}},$$
where *λ* is the depth of penetration of the electromagnetic field, *μ*_0_ is the magnetic permeability of the vacuum, γ is the electric conductivity, *μ_c_* is the magnetic permeability, and *f* is the frequency.

Numerous ready-to-use integrated circuits equipped with linear Hall-effect sensors (e.g., A1321, A1322, A1323, KSY14, HY05-P, HAS100-S, and A3515 ÷ A3518) [35] are available. The A3515LUA system, Allegro MicroSystems, Inc., Worcester, MA, USA with a Hall sensor was chosen for this project. When the Hall sensor is not affected by a magnetic field, only a constant component signal equal to about 50% of the supply voltage exists in the output of, e.g., the A3515LUA system. However, when the sensor is in the magnetic field, an additional constant or alternating component appears, depending on the direction and magnitude of the magnetic field (DC and AC).

In this paper, we focus on the modeling and practical application of the linear Hall sensor in a transducer to measure the instantaneous values of AC and DC voltage with separation output and with an open feedback loop. Simulation calculations of the main and leakage inductance for the measuring circuit were carried out with the assumption of linearity of the magnetic core. The article presents the influence of frequency on the phase-shift angle between the measured voltage and the voltage at the output of the A3515LUA system [35]. As shown in this paper, by making a small change in the primary winding of the system conditioning the output voltage signal level of the A3515LUA used in the current transducer [35], it is possible to measure the voltage’s instantaneous values. In this application, the integrated A3515LUA circuit equipped with a linear Hall-effect sensor operates in an open feedback loop [28,29,35].

In the designed voltage measurement system, an RTP ferromagnetic core was used [33]. In the air gap of the RTP ferromagnetic core, the A3515LUA system with the Hall sensor was placed. The leakage inductances of the measurement winding were determined in the Finite Element Method Magnetics (FEMM) software. Experimental investigations were carried out for the frequency of supply voltages from 0 to 25 kHz (according to the manufacturer of A3515LUA, the recommended frequency is ≤30 kHz [35]).

The rest of the paper is organized as follows: in Section 2, the system forming the voltage signal from a voltage transducer with a Hall-effect sensor is described. In Section 3, the formal analysis determination of self and leakage inductances of the measurement winding is described. Experimental investigations and practical applications of voltage transducers in electrical machine diagnostics are presented in Section 4. Finally, conclusions are developed in Section 5.

## 2. The System of a Voltage Signal Forming from a Voltage Transducer with a Hall-Effect Sensor

The RTP symbol 26.9x14.5x11-52 (manufactured by Feryster Company, Ilowa, Poland) [33] ferromagnetic core was chosen for the implementation of the voltage transducer. Figure 1 presents the designations of the most important electrical quantities and construction parameters describing the RTP ferromagnetic core with the δ air gap and the measuring winding with the number of windings being *N*_1_ = 1500. The value of the rated current magnetic fluxing through the measurement winding was *I*_1N_ = 10 mA for the measured rated voltage *U_X_*_N_, as it is in voltage transducers, e.g., those manufactured by LEM (LEM International SA, Chemin des Aulx, Switzerland) [4]. The A3515LUA integrated circuit in the air gap of the core has a housing thickness of 1.47–1.57 mm [35]. Therefore, in our tests, the air gap changed from 1.6 to 1.8 mm. For the air gap δ ∈ {1.6–1.8 mm}, and for the magnetic permeability of the RTP core (according to the manufacturer specification, it is equal to μ*_c_* ∈ {15–100}) and core conductivity of γ = 0.00869 MS, some tests were run [33]. Moreover, Figure 1b shows the basic operating system of the voltage transducer in the open feedback loop, which shaped the voltage signal from the A3515LUA Hall sensor output proportional to the measured voltage *u_x_* (AC and DC).

The presented voltage transducer (see Figure 1) with the A3515LUA system requires only a stabilized 5-V supply voltage [35]. In the system shown in Figure 1b, two operational amplifiers in US1 eliminate the DC voltage component (which, depending on the A3515LUA system for *u_x_* = 0 V, ranges from 2.425 V to 2.575 V [35]). According to the manufacturer, the voltage sensitivity at the output of the A3515LUA Hall sensor is *k_H_* = 50 V/T or 5 mV/Gauss [35]. A potentiometer (20 kΩ) is used to eliminate the constant component from the voltage *u*_1_ (from the A3515LUA output). Two operational amplifiers in US2 convert voltage *u*_1_ to current *i*_2_. Current *i*_2_, at the output of US2, is converted to voltage *u*_2_ under the resistance *R* = 1 kΩ, and *u*_2_ is proportional to the measured current *i*_1_ (or voltage *u_x_*). The gain of the transducer (see Figure 1) was chosen so that the amplitude of the rated current *I*_m1N_ at 50 Hz could give an amplitude of the voltage at the output of the transducer of *U*_m2*N*_ = 10 V.

The following factors were introduced to the circuit by the measuring system (see Figure 1) for AC voltages:An additional impedance in which the part associated with the inductive reactance introduces a phase-shift angle between the measured voltage *u_x_* and the current *i*_1_ (see Figure 1);A phase-shift angle due to the relationship between the magnetic flux density *B* and the magnetic field strength *H* (which, in modeling for AC signals, is most often taken into account by the real and imaginary parts of the magnetic permeability *μ*; see Equation (1)).

## 3. Determination of Self and Leakage Inductances of the Measurement Winding

### 3.1. Formal Analysis

For low frequencies (for which the skin effect and hysteresis can be neglected in the core) and assuming simplification of the straight-line distribution of magnetic field lines in the core’s air gap, the flux in the core is equal to that in the air gap, *B*_c_*A*_c_ = *B_δ_A_δ_* and substituting *H_c_* = *B_c_*/(*μ*_0_*μ*_c_), the magnetic flux density *B_c_* of the magnetic field in the core (see Figure 1) is as follows [36,37]:(3)$$B_{c}=\mu_{0}\frac{i_{1}N_{1}}{\frac{A_{c}}{A_{\delta }}\delta +\frac{\left(2\pi -\alpha \right)r}{\mu_{c}}},$$
where *A_c_* is the cross-sectional area of the core, *A_c_* = *ab* (see Figure 1), *B_δ_* is the magnetic flux density in the air gap, *A_δ_* is the cross-sectional area of the air gap, *δ* is the length of the air gap, *H_c_* is the magnetic field strength in the core, *µ*_0_ is the magnetic permeability of free space, *µ*_c_ is the relative core magnetic permeability, and *α* is the angle (see Figure 1).

Assuming simplification for small angles *α* (see Figure 1), *α* = 2arcsin(*δ*/2*r*) ≈ *δ*/*r*, and the magnetic flux density in the core is equal to(4)$$B_{c}=\mu_{0}\frac{i_{1}N_{1}}{\frac{A_{c}}{A_{\delta }}\delta +\frac{2\pi r-\delta }{\mu_{c}}}.$$

Sometimes, for a core with an air gap, instead of a relative magnetic permeability *μ_c_*, the effective relative permeability *μ_ce_* is used as follows:(5)$$\mu_{ce}=\frac{l_{c}+\delta }{\delta +\frac{l_{c}}{\mu_{c}}}=\mu_{c}\frac{1+\delta }{1+\mu_{c}\frac{\delta }{l_{c}}},$$
where *l_c_* is the average core length (see Figure 1a).

To calculate the inductance of the measurement winding (see Figure 1), the distribution of the magnetic flux in the core with the air gap should be determined. Assuming that the elementary magnetic flux d*Φ* passing through the elementary surface d*A* is equal to d*Φ* = *B_c_*d*A*, and including Equation (4), the magnetic flux in the core with the air gap is(6)$$\Phi =b\int_{r_{1}}^{r_{2}}\mu_{0}\frac{i_{1}N_{1}}{\frac{A_{c}}{A_{\delta }}\delta +\frac{\left(2\pi -\alpha \right)r}{\mu_{c}}}dr.$$

Considering that the total magnetic flux in the core is *N*_1_ times larger, after the integration of Equation (6), we get(7)$$\Psi =\frac{\mu_{c}\mu_{0}bN_{1}^{2}i_{1}}{\left(2\pi -\alpha \right)}ln\frac{1+\frac{A_{\delta }}{A_{c}}\frac{\left(2\pi -\alpha \right)r_{2}}{\mu_{c}\delta }}{1+\frac{A_{\delta }}{A_{c}}\frac{\left(2\pi -\alpha \right)r_{1}}{\mu_{c}\delta }}.$$

The main inductance of the measuring winding associated with the main magnetic flux in the core length *l_c_* with the air gap *δ* (see Figure 1) can be defined as(8)$$L_{c}=\frac{\Psi }{i_{1}}=\frac{\mu_{c}\mu_{0}bN_{1}^{2}}{\left(2\pi -\alpha \right)}ln\frac{1+\frac{A_{\delta }}{A_{c}}\frac{\left(2\pi -\alpha \right)r_{2}}{\mu_{c}\delta }}{1+\frac{A_{\delta }}{A_{c}}\frac{\left(2\pi -\alpha \right)r_{1}}{\mu_{c}\delta }}.$$

The total inductance *L* is the sum of the inductance of the winding associated with the main *L_c_* magnetic flux, and the leakage inductance *L_σ_* associated with the leakage magnetic flux, which is *L* = *L_c_* + *L*_σ_.

Figure 2 shows the effect of the magnetic permeability of the RTP, symbol 26.9 × 14.5 × 11 − 52, which is *μ_c_* ∈ {15–100} at γ = 0, during the inductance changes of the measuring winding *L_c_* (see Figure 2) for the air gap varying in the range *δ* ∈ {1.6–1.8 mm} and for a frequency of *f* = 0 Hz (i.e., not affected by the skin effect). In the calculations, it was assumed that the equivalent cross-sectional area of the magnetic flux in the air gap was *A_δ_* = (*a* + 2*δ*)(*b* + 2*δ*) [5].

Calculations of inductance for alternating voltages at frequencies in kHz and above are very complex due to eddy currents and the skin effect, as well as phase shift caused by the relationship between the magnetic flux density *B* and the magnetic field strength *H*. Even the modeling of the field distribution in the finite element method software is imperfect, and each attempt to create mathematical models must be verified experimentally [32,36].

### 3.2. Simulations of Leakage Inductance of Measuring Winding Using FEMM

The leakage inductance of the measuring winding was determined in two dimensions (2D) in the Finite Element Method Magnetics (FEMM) program [38]. For this purpose, the geometry of the transducer model with the measuring winding was created. To determine the leakage inductance of the measurement winding, a second auxiliary compensating winding (wound on an RTP 26.9 × 14.5 × 11 mm core) was introduced into the model, and supplied with the same current *I*_1_ as in the measuring winding. The simulations were carried out for *N*_1_ = *N*_2_ = 1500 turns at *I*_1_ = *I*_2_ = 0.01 A. In Figure 3, for the compensated magnetic flux condition *I*_1_*N*_1_ = *I*_2_*N*_2_, a simplified three-dimensional (3D) view of the RTP core with two windings and a 2D view of the resultant distribution of the magnetic field in the ferromagnetic core (RTP 26.9 × 14.5 × 11) in the FEMM software are presented.

The results of the simulations showed that the leakage inductances of the measurement winding *L*_σ1_ and the auxiliary winding *L*_σ2_ lying side by side (Figure 3) are equal to *L*_σ1_ = *L*_σ1_ = *L*_σ_ = 49.7 μH. If, on the RTP core, one winding is over the other, the inductance of the measurement winding closest to the core (Figure 3) is equal to *L*_σ1_ = 49.7 μH, and that of the winding lying outside *L*_σ2d_ is 505 μH. In the case of a voltage transducer with a feedback loop, both leakage inductances will add up and increase the phase-shift angle between the measured voltage *U_x_* and the voltage *U*_2_ (see Figure 1).

The influence of leakage inductance *L*_σ_ of the measuring winding is small; for example, for a 30 kHz supply voltage, the leakage reactance is equal to *X*_σ_ = 9.37 Ω. The resistance of the measuring winding is approximately 200 Ω. With direct voltage measurement, where *R_d_* = 0 Ω (see Figure 1), the phase-shift angle γ_σ_ = 2.7° is given. In the case of *U_x_* = 100 V and *R_d_* = 10 kΩ, the leakage reactance of the measurement winding for the 30 kHz of supply voltage gives a phase-shift angle of γ_σ_ = 0.05°.

## 4. Experimental Investigations

### 4.1. Laboratory Tests

Figure 4 shows the test bench and the view of constructed voltage transducers containing RTP cores with A3515LUA Hall sensors. The authors would like to mention that 60 voltage transducers (the same as discussed in this paper) and 60 current transducers (discussed in Reference [29]) were assembled on eight laboratory test benches in the Laboratory of Electrical Machines of the Kielce University of Technology.

The influence of the supply voltage frequency on the shape and phase shift of the measured voltages was carried out using a purpose-built power amplifier, working on two integrated circuits (STK4050II). Two integrated circuits (STK4050II) were combined into a so-called bridge system to obtain a voltage amplitude (at the output of the power amplifier) that was twice as high with respect to the supply voltages.

The test bench (Figure 4) consisted of the following devices:The power amplifier operating in the bridge configuration with two integrated circuits STK4050II;Two power supplies with adjustable voltages of 0–360 V;A Hamag HMF 2525 generator with adjustable frequency and amplitude sine signals;A set of constructed transducers for voltage measurement (and currents);A Tektronix MSO 3014 digital oscilloscope.

Sinusoidal voltage signals from the HMF 2525 generator with frequencies from 0 to 25 kHz were supplied to the power amplifier (working on two STK4050II systems). The power amplifier load was a reference resistor of *R*_r_ = 1000 Ω of 0.01% class with a 1:3 voltage divider. The tests were carried out at a constant voltage amplitude of *U*_mx_ = 60 V at the output of the amplifier (measured on the reference resistor *R*_r_). The shapes of the supply voltage and voltages from the two transducers with *R_d_*_1_ = 5.6 kΩ and *R_d_*_2_ = 22 kΩ were recorded using the MSO 3014 digital oscilloscope. Figure 5 shows the recorded waveforms of the measured voltage at 100 Hz, 1 kHz, 10 kHz, and 25 kHz. The waveforms were recorded using a four-channel TDS 2024 digital oscilloscope (Tektronix) under steady-state conditions. The four signals were connected to particular channels of the digital oscilloscope MSO 3014:Channel 1—open-loop transducer (see Figure 1) with *R_d_* = 5.6 kΩ;Channel 2—open-loop transducer (see Figure 1) with *R_d_* = 22 kΩ;Channel 3—voltage transducer with compensation winding (LV 25-P manufactured by LEM with *R_d_* = 5.6 kΩ), an overall accuracy of ±0.9% (primary nominal root mean square (RMS) current *I_PN_* = 10 mA, *T_air_* = 25 °C, and supply voltage ±15 V), a linearity error of <0.2%, and a step response time of 40 µs to 90% of *I_PN_* (with *R_d_* = 25 kΩ) [10];Channel 4—passive voltage probe P2220 (Tektronix) with an attenuation ratio of 1:1 and a DC bandwidth to 200 MHz. The probe was connected to the output of the power amplifier (reference voltage signal) and to the reference resistor with the 1:3 divider.

The recorded voltage waveforms (see Figure 5) and those for the 100-Hz voltages were subjected to Fourier analysis. In Figure 6, the comparisons of the higher harmonic contents (*IHD*) at voltages with frequencies of 100 Hz, 1 kHz, 10 kHz, and 25 kHz at the output of the amplifier and on the output of the open-loop voltage transducers with *R_d_*_1_ = 5.6 kΩ and *R_d_*_2_ = 22 kΩ are presented. The higher harmonic contents (*IHD*) are compared with the fundamental voltage component (at the output of the amplifier and at the output of the tested transducers).

Table 1 presents a comparison of the total harmonic distortion (*THDu*) shown in Figure 6 (i.e., for the fundamental frequencies *f* ∈ {100 Hz, 1 kHz, 10 kHz, 25 kHz}) and additionally for the fundamental frequencies *f* ∈ {500 Hz, 5 kHz, 15 kHz}. The *THDu* was calculated up to the 39th harmonic according to the following equation [39]:(9)$$THDu=\frac{\sqrt{\sum_{v=2}^{k}U_{mv}^{2}}}{U_{m1}}100,$$
where *U_mv_* is the the amplitude of the *v*-th harmonic, and *U_m_*_1_ is the the amplitude of the fundamental component.

In the *THDu* calculations presented in Table 1, the following designations were introduced:*THDu_x_*—in the voltage at the output of the amplifier (the reference voltage);*THDu_Rd_*_1_—in the voltage at the transducer output (see Figure 1) with *R_d_*_1_ = 5.6 kΩ;*THDu_Rd_*_2_—in the voltage at the transducer output (see Figure 1) with *R_d_*_2_ = 22 kΩ.

Phase-shift angle calculations were made for the fundamental components of the recorded waveforms obtained due to Fourier analysis. The results of the phase-shift angles are given in Table 2.

Figure 7 shows phase shifts and amplitudes of signals for open-loop transducers with *R_d_* ∈ {5.6 kΩ, 22 kΩ} and closed-loop transducers (LEM) with *R_d_* ∈ {5.6 kΩ, 22 kΩ} as a frequency function (given in Table 2).

From the recorded waveforms (see Figure 7), it can be determined that, for an increasing frequency and the same amplitude of the measured voltage, the following is true:The voltage amplitudes at the output of the tested transducers decrease. This is to a small extent due to the increase in the reactance of the measurement circuit that results in the reduction of the current *I*_1_ and, above all, the reduction of magnetic flux density in the core (and in the air gap where Hall sensor A3515LUA is located) caused by the phenomenon of the epidermal magnetic field in the core;The phase-shift angle between the measured voltage and the voltage at the output from the Hall sensor A3515LUA increases, which is affected by both the influence of the measuring circuit reactance and the phase shift between the magnetic flux density *B* and the magnetic field strength *H*, causing the angle of the phase shift to increase above 180°.

Increasing the resistance of the additional measuring circuit allows for the DC component and low frequencies to increase the range of voltage measurement (e.g., from *R_d_*_1_ = 5.6 kΩ to R_d2_ = 22 kΩ, an increase of four times). At the same time, for the same amplitude of the AC component, it causes the following:A reduction of the phase-shift angle between the measured voltage and the output voltage of the transducer (see Figure 5 and Figure 7);An increase of more than twice in the range of 1–10 kHz *THDu* (Table 1);Smaller amplitude changes in the range of 2–15 kHz in the tested transducer without feedback than in the transducer with feedback (LEM);On the one hand, an improvement in the quotient of the resistance to the reactance of the measuring circuit, but, on the other hand, a decrease in the value of the current of the measuring winding, which results in additional noise and distortion of the measurement signal (see Figure 5a). The maximum frequency of the measured voltage and the practical application of the designed open-loop transducer is 20 kHz (see Figure 5—channels 1 and 2), and, up to this frequency, amplitudes at the output and its waveform are the most similar to the reference voltage (see Figure 5—channel 4).

### 4.2. Practical Utilization of Voltage Transducers in Electrical Machine Diagnostics

After the research and tests aimed at determining the most important metrological properties of the proposed transducers, we focused on their operation in monitoring or protection systems of electrical machines and devices. It appears to be particularly important to verify the suitability of these properties in algorithms, where three or more phase voltages or phase-to-phase voltages must be measured and simultaneously processed. A frequently used algorithm for processing measured signals involves determining the asymmetry of supply voltages on the basis of symmetrical components [5]. For this analysis, simultaneous measurement of three phase voltages is required. The measurement of the voltages should not be affected by the error caused by the angular shift between samples taken at the same time from different phases. It is also very important to determine the contents of higher harmonics of the measured voltages. Practical utilization of voltage transducers are presented on registered voltages in electrical machine diagnostics in the transducer example (see Figure 1) with 22 kΩ, and in the transducer made by LEM with 22 kΩ.

Presented waveforms after correction are obtained due to the Fourier decomposition and then for the *v*-th harmonic using multipliers amplitude *k_v_* and phase *β_v_* corrections (see Figure 7). The waveform of voltage after correction *u_after_cor_* can be expressed as(10)$$u_{after\_cor}=U_{0}+\overset{n}{\underset{v=1}{\sum }}k_{v}U_{mv}\sin(v\omega t+\phi_{v}+\beta_{v}),$$
where *U_mv_* and *ϕ_v_* are the amplitude and phase angle of the *v*-th harmonic due to the Fourier decomposition, respectively, *U*_0_ is the constant component, *ω* is the electrical angular velocity, *t* is the time, *k_v_* is the amplitude correction factor (defined for a particular harmonic as a difference between the number 2 and a value obtained from amplitude characteristic from Figure 7), and *β_v_* is the phase correction factor (defined for a particular harmonic as a value obtained for phase characteristic from Figure 7 and Equation (10) into which it is put with a minus sign).

In order to simplify the voltage calculations for a big numer of harmonics, amplitude–phase characteristics for any transducer should be approximated by a function which is the most suitable for that. In this paper, third-order polynomial functions were used.

In Figure 8, *u_a_*, *u_b_*, and *u_c_* waveforms of the recorded phase-to-neutral voltages in the armature stator windings of the 5.5-kVA salient pole synchronous generator are presented. Figure 9 presents a comparison of the higher harmonic contents (*IHDu*) obtained due to Fourier analysis and the phase-established difference of individual harmonics (the asymmetry diagnostics) for the fundamental, third, fifth, and seventh harmonics in supply armature voltages of the synchronous generator at 5.5 kVA recorded by the examined voltage transducers (see Figure 4). The voltages were recorded in the Laboratory of Electric Machines at Kielce University of Technologies, and the harmonics decreased by the amplitude factor shown in Figure 7. Registered waveforms from the transducer (Figure 1) with 22 kΩ and the transducer made by LEM with 22 kΩ were the same as in Figure 8, and that is why they are not presented (in Figure 8). A detailed analysis is given by comparison of the higher harmonic contents in Figure 9 and in Table 3 and Table 4.

In the voltage waveforms (see Figure 8a), the participation of higher harmonic contents from the 11th harmonic to the 40th harmonic is less than 0.2% (with respect to the fundamental component), and their contribution is very small. Total harmonic distortions (*THD*) from Equation (9) in voltages (see Figure 8) were obtained due to Fourier analysis up to the 39th harmonic [39] with and without correction, and the errors are given in Table 4. The errors were determined with respect to the *THD* of the passive voltage probe.

In Figure 9, the fundamental and distorted line-to-line voltages for the output voltage inverter are presented. The voltages were recorded in the Laboratory of Electric Machines at Kielce University of Technologies during the supply of a squirrel cage induction motor by a voltage inverter. Registered waveforms by the passive voltage probe, transducer (see Figure 1) with 22 kΩ, and transducer made by LEM with 22 kΩ with and without correction are presented in Figure 10.

In Figure 10, the comparison of the higher harmonic contents (*IHD_u_*) obtained due to Fourier analysis in the waveforms of line-to-line voltages for the voltage inverter registered by the passive voltage probe from 0 to 25 kHz are presented.

In Figure 11, waveforms of the line-to-line voltages for the output voltage inverter registered by the voltage probe *U_ab_*, the transducer (see Figure 1) with 22 kΩ, and the transducer made by LEM [24] with 22 kΩ with and without correction are presented. For better visibility in Figure 11a,b, the waveforms with and without correction are presented for a 8-ms time span.

The *THD* from Equation (9) in voltages (see Figure 11) was obtained due to Fourier analysis up to 500 harmonics with and without correction, and the errors are given in Table 5. The errors were determined with respect to the THD of the passive voltage probe.

From the comparison of *THD* in Table 5, it follows that, in distorted waveforms of the line-to-line voltages (up to the 500th harmonic), the transducers with both open loop (see Figure 1 with correction) and closed feedback loop (made by LEM with correction) provide similar measured values with respect to the passive voltage probe (Tektronix).

## 5. Conclusions

This paper presented a design of a DC and AC voltage transducer without a feedback loop that measures voltages of frequency up to 25 kHz. The application of a transducer without the feedback loop to measure phase-to-phase voltages in electric machines with lower-order harmonics and line-to-line voltages in output voltage inverters with higher-order harmonics up to 25 kHz was proposed. On the basis of the experiments, it can be stated that, in the case of voltage measurements up to 300 Hz, the effects of main reactance, the leakage of the measurement winding, and the core parameters are very small and are, therefore, negligible. After entering the correction of the amplitude and phase angle, the transducers with both open and closed feedback loops have similar measuring properties to the voltage probe. For a frequency of 1 kHz, the phase-shift angle between the measured voltage and the output voltage from the Hall sensor A3515LUA is already over 10°; however, at frequencies above 7 kHz, it exceeds 90°.

The comparison of the designed voltage transducer with the LEM transducer showed that, at frequencies above 7 kHz, the effect of hysteresis on the phase-shift angle between the magnetic flux density *B* and the magnetic field strength *H* increases, which results in a significant increase in the phase-shift angle, since, in the closed-loop voltage transducer (LEM), there is only a residual magnetic flux in the core and the phase shift is not affected by the magnetic hysteresis. In these transducers (LEM), the phase-shift angle does not increase by more than 73° within the tested range of up to 25 kHz.

If the phase-shift angle is of no importance in voltage measurements, and only the shape of the voltage waveform and its instantaneous values in the frequency range up to 1 kHz are valid, the open-loop system with the A3515LUA sensor can be used. In addition, in the frequency range of 15–25 kHz, the tested transducers have a drop in voltage amplitude similar to that shown by the factory transducer (LEM) with a closed feedback loop. To apply the voltage transducer without a feedback loop (or the factory transducer made by the LEM Company with a closed loop) for measurements up to 25 kHz (up to 500 harmonics), in this article, amplitude and phase-angle correction were applied.

In the measurement of voltages in a wide range of frequencies, knowledge of the amplitude–phase characteristics of a voltage transducer are required. Thanks to the amplitude and phase-angle correction, the constructed transducers can be used in supply voltage measurement, monitoring, and in the protection systems of electrical machines and voltage inverters. For this reason, it was decided that this method should be described in this paper, especially where the cost of the transducer is relatively low and is about 9 United States dollars (USD) and the method allows for voltage measurement in electrical systems containing voltage inverters.

Measured voltages and the processed information concerning the content of higher harmonics using the method of correction presented in the work can be used, for example, in diagnostics or security systems intended for a given type of electrical machine and voltage inverter. The disadvantage of voltage measurement using the method described in this paper is the fact that this is a contact method and it requires the knowledge of amplitude–phase characteristics of the selected transducer, as well as an approximation of these characteristics, and amplitude and phase error correction.

## Figures and Tables

**Figure 1 sensors-19-01071-f001:**
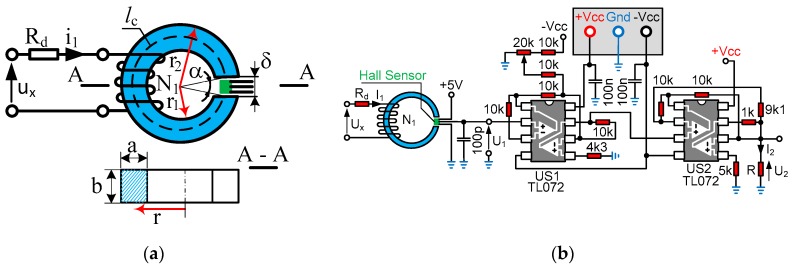
The shaping system of the A3515LUA voltage signal from the Hall sensor A3515LUA output in a system with no feedback loops: (**a**) measuring circuit with a cross-section of the toroidal core; (**b**) simplified basic transducer working system.

**Figure 2 sensors-19-01071-f002:**
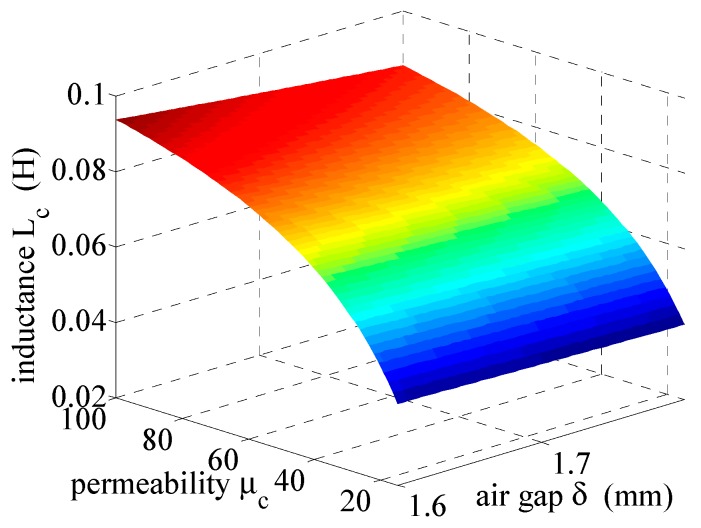
Effect of magnetic permeability *μ_c_* ∈ {15–100} and air gap *δ* ∈ {1.6–1.8 mm} at an electric conductivity of γ = 0 on inductance changes of the measuring winding *L_c_*.

**Figure 3 sensors-19-01071-f003:**
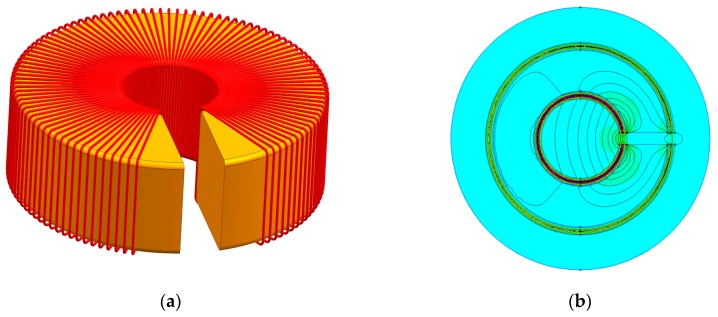
View of (**a**) the measuring and auxiliary compensating windings in three dimensions (3D) on the core with the air gap; (**b**) the magnetic flux distribution in two dimensions (2D) when the measuring winding and auxiliary winding (*I*_1_*N*_1_ = *I*_2_*N*_2_) are simultaneously fed.

**Figure 4 sensors-19-01071-f004:**
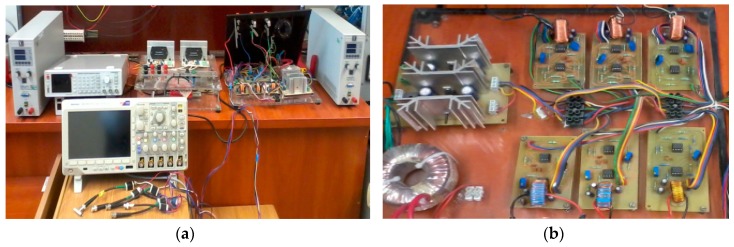
Test bench: (**a**) general view; (**b**) view of three constructed voltage transducers (at the top) and three current transducers (at the bottom) in the Laboratory of Electrical Machines of the Kielce University of Technology.

**Figure 5 sensors-19-01071-f005:**
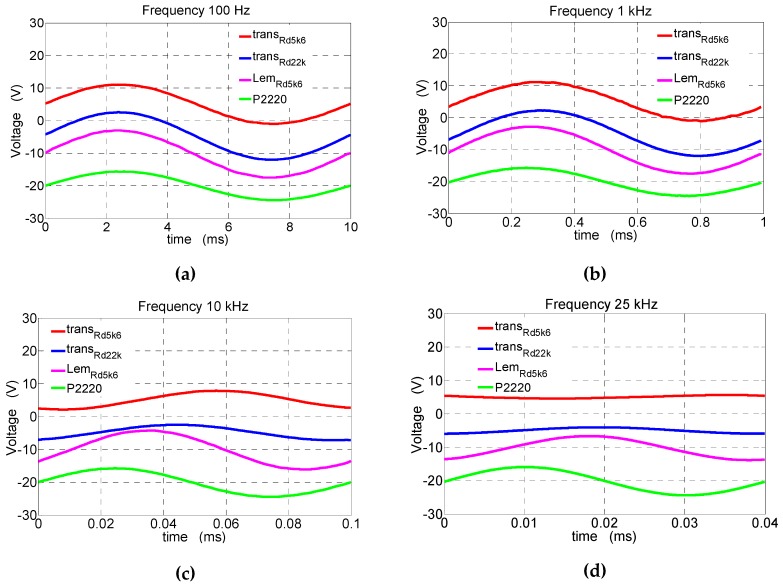
Voltage waveforms recorded by the designed transducers, and the reference signal waveforms recorded using the passive voltage probe P2220 for frequencies (**a**) 100 Hz, (**b**) 1 kHz, (**c**) 10 kHz, and (**d**) 25 kHz, (abbreviations: trans_Rd5k6_—open-loop voltage transducers with Rd1 = 5.6 kΩ; trans_Rd22k_—open-loop voltage transducers with Rd1 = 22 kΩ; Lem_Rd5k6_—voltage transducer with compensation winding with Rd1 = 5.6 kΩ; P2220—passive voltage probe).

**Figure 6 sensors-19-01071-f006:**
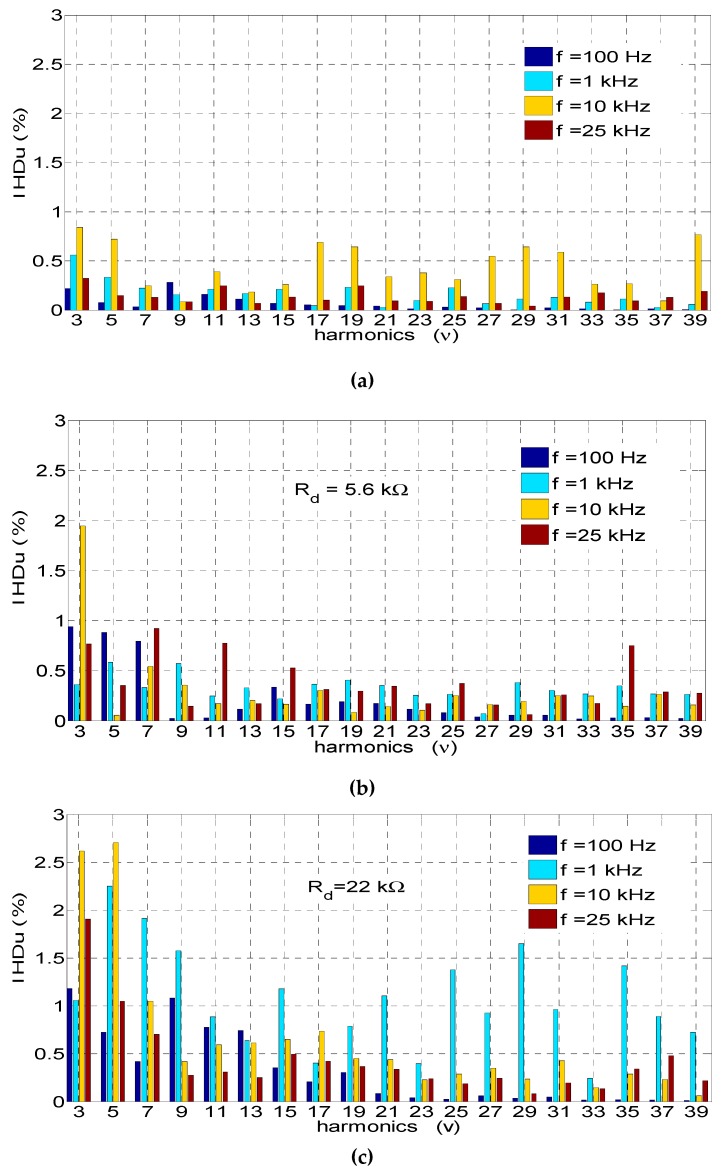
Comparison of higher harmonic contents (up to 39 harmonics) in measured voltages for frequencies of 100 Hz, 1 kHz, 10 kHz, and 25 kHz on the outputs of (**a**) the amplifier with the reference signal, (**b**) the open-loop transducer with *R_d_*_1_ = 5.6 kΩ, and (**c**) the open-loop transducer with *R_d_*_2_ = 22 kΩ.

**Figure 7 sensors-19-01071-f007:**
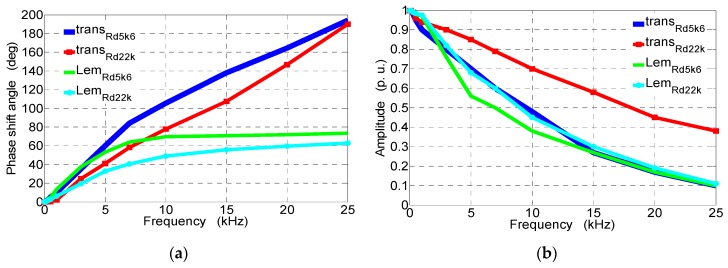
Experimental study characteristics of the open-loop transducers *R_d_* ∈ {5.6 kΩ, 22 kΩ} (see Figure 1) and the closed-loop transduceres (made by LEM) *R_d_* ∈ {5.6 kΩ, 22 kΩ}: (**a**) phase-shift angle relative to the reference voltage signal (channel 4); (**b**) amplitude of the output signals in p.u.

**Figure 8 sensors-19-01071-f008:**
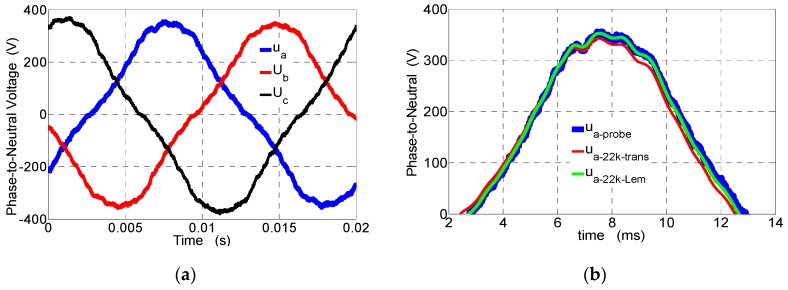
Waveforms of the stator phase-to-neutral voltages for the 5.5-kVA salient pole synchronous generator measured by (**a**) voltage probes, (**b**) a voltage probe, a transducer (see Figure 1) with 22 kΩ, and a transducer made by LEM with 22 kΩ (without correction), and (**c**) a voltage probe, a transducer (see Figure 1) with 22 kΩ, and a transducer made by LEM with 22 kΩ for the third and fifth harmonics after correction.

**Figure 9 sensors-19-01071-f009:**
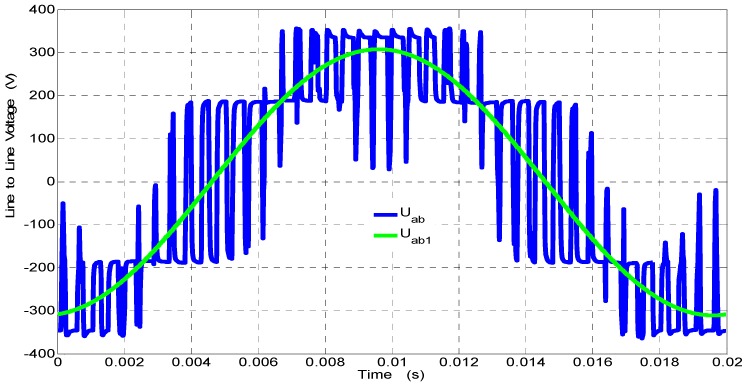
Waveforms of the fundamental and distorted line-to-line voltages for the output voltage inverter.

**Figure 10 sensors-19-01071-f010:**
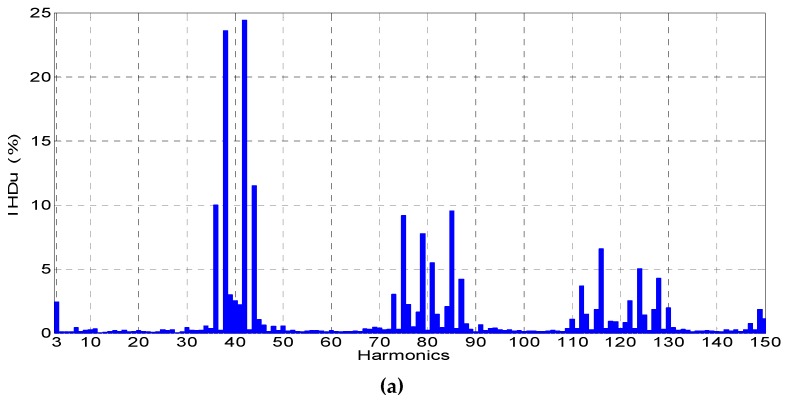
Comparison of the higher harmonic contents (*IHD_u_*) in the waveforms of line-to-line voltages for the voltage inverter registered by the voltage probe (**a**) from 0 to 7.5 kHz, (**b**) from 7.5 to 15 kHz, and (**c**) from 15 to 25 kHz.

**Figure 11 sensors-19-01071-f011:**
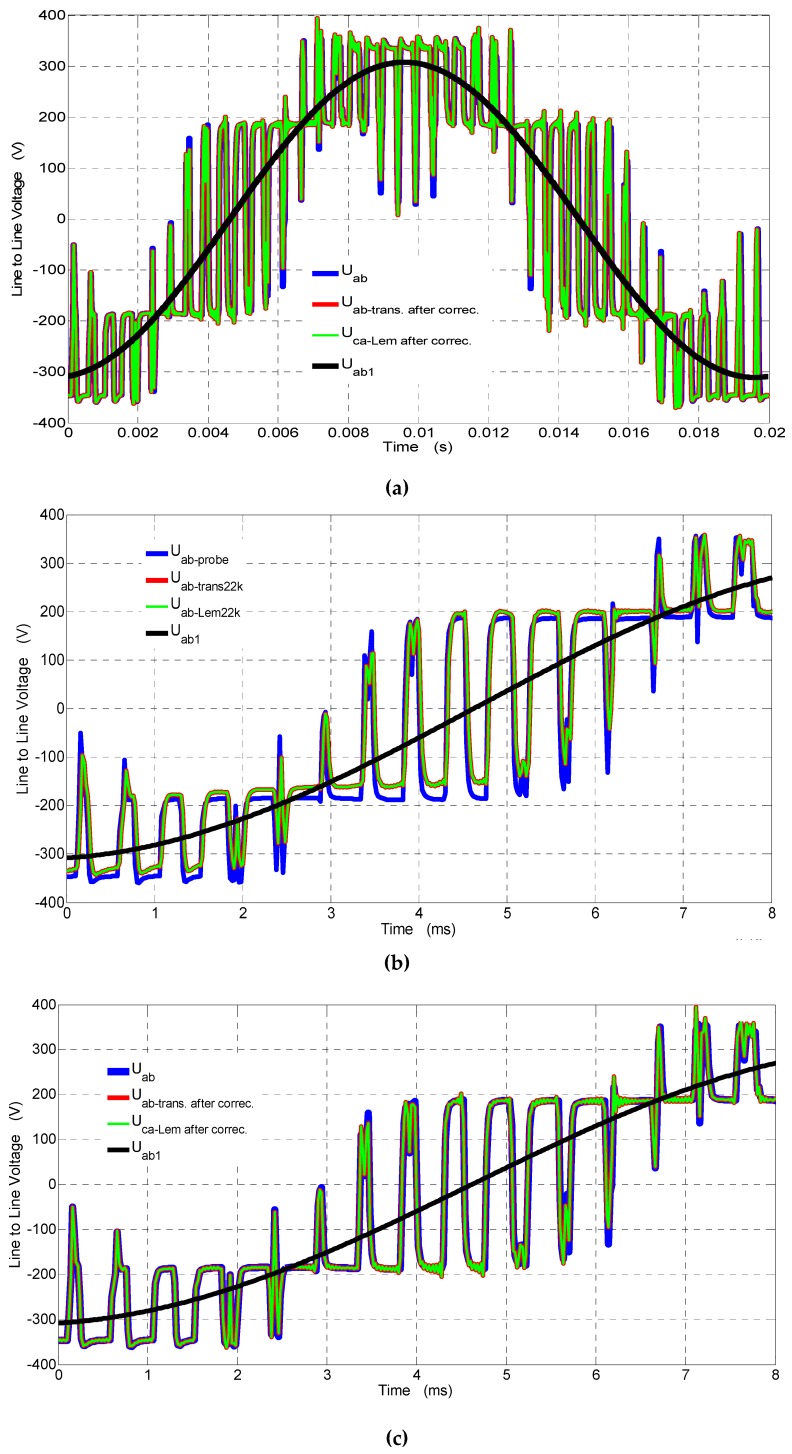
Waveforms of the line-to-line voltages for the output voltage inverter measured by (**a**) the voltage probe *U_ab_*, the transducer (see Figure 1) with 22 kΩ after correction (*U_ab_*_-*trans. after correc.*_), and the transducer made by LEM with 22 kΩ after correction (*U_ab_*_-*Lem after correc*._) for 20 ms; (**b**) the voltage probe *U_ab_*, the transducer (see Figure 1) with 22 kΩ before correction (*U_ab_*_-*trans*22*k*_), and the transducer made by LEM with 22 kΩ before correction (*U_ab_*_-*trans*22*k*_) for 8 ms; and (**c**) the voltage probe *U_ab_*, the transducer (see Figure 1) with 22 kΩ after correction (*U_ab_*_-*trans. after correc.*_), and the transducer made by LEM with 22 kΩ after correction (*U_ab_*_-*Lem after correc*._) for 8 ms.

**Table 1 sensors-19-01071-t001:** Total harmonic distortion (*THDu*) for frequencies *f* ∈ {100 Hz, 500 Hz, 1 kHz, 5 kHz, 10 kHz, 15 kHz, 20 kHz, and 25 kHz}.

*f* (Hz)	*THDu_x_* (%)	*THDu_Rd_*_1_ *R_d_*_1_ = 5.6 kΩ (%)	*THDu_Rd_*_2_ *R_d_*_2_ = 22 kΩ (%)
100	0.43	1.59	2.16
500	0.89	1.89	4.85
1000	0.88	1.49	5.17
5000	1.34	1.37	4.03
10,000	2.14	2.19	4.26
15,000	1.82	2.29	3.73
20,000	1.77	3.49	2.97
25,000	0.67	1.94	2.59

**Table 2 sensors-19-01071-t002:** The angles of the phase shift between the reference voltage signal (channel 4) and the voltages from the open-loop transducers and those with the closed loop (LEM) with *R_d_* ∈ {5.6 kΩ, 22 kΩ}.

Frequency (Hz)	Phase shift angle for different *R_d_* (°)				Amplitude for different *R_d_* (p.u.)			
	Figure 1 5.6 kΩ	Figure 1 22 kΩ	LEM 5.6 kΩ	LEM 22 kΩ	Figure 1 5.6 kΩ	Figure 1 22 kΩ	LEM 5.6 kΩ	LEM 22 kΩ
100	~0.1	~0	~0.1	~0.1	1.00	1.00	1.00	1.000
300	2.6	0.2	3.7	1.5	0.98	0.99	0.99	0.990
500	5.1	0.4	4.7	2.5	0.95	0.96	0.98	0.985
1000	10.2	2.1	13.4	6.2	0.90	0.94	0.97	0.975
3000	35.3	24.7	37.4	19.4	0.80	0.9	0.76	0.820
5000	60.1	40.9	53.3	32.8	0.70	0.85	0.56	0.680
7000	84.0	58.2	64.0	40.5	0.6	0.79	0.50	0.600
10,000	105.5	77.7	69.5	48.9	0.48	0.70	0.38	0.450
15,000	137.9	107.6	70.8	55.6	0.27	0.60	0.27	0.300
20,000	164.4	146.7	71.9	59.5	0.17	0.45	0.17	0.190
25,000	194.2	190.1	73.3	62.8	0.10	0.38	0.10	0.110

**Table 3 sensors-19-01071-t003:** Higher harmonic contents in *U_a_* phase-to-neutral voltages (see Figure 8) obtained due to Fourier analysis in relation to the fundamental harmonic and probe harmonic phase angles.

Harmonic	Probe	Transducer (Figure 1) R_d_ = 22 kΩ		Transducer (Figure 1) after correction R_d_ = 22 kΩ		Transducer (LEM) R_d_ = 22 kΩ		Transducer (LEM) after correction R_d_ = 22 kΩ	
(k)	(%)	Amplitude (%)	Phase shift (°)	Amplitude (%)	Phase shift (°)	Amplitude (%)	Phase shift (°)	Amplitude (%)	Phase shift (°)
3	11.33	11.22	1.19	11.32	0.29	11.31	2.79	11.33	0.23
5	2.65	2.59	24.67	2.64	0.56	2.64	20.64	2.65	0.51
7	1.60	1.57	56.37	1.60	0.79	1.58	40.16	1.60	0.78
9	0.47	0.45	109.56	0.47	0.91	0.46	56.23	0.47	0.90

**Table 4 sensors-19-01071-t004:** Total harmonic distortion (*THD*) from Equation (9) in voltages (see Figure 8) obtained due to Fourier analysis up to the 39th harmonic.

-	Passive voltage probe	Transducer (Figure 1) R_d_ = 22 kΩ	Transducer (Figure 1) after correction R_d_ = 22 kΩ	Transducer (LEM) R_d_ = 22 kΩ	Transducer (LEM) after correction R_d_ = 22 kΩ
***THD* (%)**	12.13	11.81	12.11	11.96	12.12
***Error* (%)**	-	2.64	0.16	1.40	0.08

**Table 5 sensors-19-01071-t005:** Total harmonic distortion (*THD*) from Equation (9) in voltages (see Figure 8) obtained due to Fourier analysis up to 500 harmonics.

-	Passive voltage probe	Transducer (Figure 1) R_d_ = 22 kΩ	Transducer (Figure 1) after correction R_d_ = 22 kΩ	Transducer (LEM) R_d_ = 22 kΩ	Transducer (LEM) after correction R_d_ = 22 kΩ
***THD* (%)**	45.93	41.34	45.87	40.16	45.55
***Error* (%)**	-	9.99	0.13	12.78	0.83

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
