# Peer review of "Practical Adaptation of a Low-Cost Voltage Transducer with an Open Feedback Loop for Precise Measurement of Distorted Voltages"

_sensors, 2019, doi:10.3390/s19051071_

Round 1

Reviewer 1 Report

Author present a practical study to use low-cost voltage transducers to measure distorted voltages. The paper is in general well written an easy to read, and the results are of interest. Some of my general concerns:

A) I think authors should clearly mention the advantages of the proposed method compared to other methods presented in the literature.

B) Some parts of the paper need to be written more carefully (see comment 2 and 11, for example).

Some specific comments for authors:

1) Line 94: I don’t understand the “However, ...”. Why however?

2) The end of the introduction is a bit confusing. I think this part of the paper should not include that amount of details.

3) Section 2: I think many of the data presented here has been already introduced.

4) I think authors should clearly mention their contribution compared to other references.

5) Page 4, “An additional impedance Z in which...”: Please mention why is this included.

6) Section 3.1: Please include some references to see where the basic theory has been taken from.

7) After (6), “...time larger, after the integration (5), we get:”: I think it should be “(6)”.

8) Figure 5, (b): I suggest using the same number of cycles of Figure 5, a, c, and d.

9) Line 303: This (2) should be a [2]?

10) Figure 11: I think the lines are very thick. I suggest using thinner lines.

11) I think equation 10 should be included before presenting the results, or in an Appendix.

12) I think the last sentences of the conclusion should be included together.

13) I think it would be a good idea to mention what would be the reduction in price if the proposed system is used (compared to LEM).

14) Can authors give the values of components in Figure 1 (b)? Also, can authors mention why they need a capacitor after the hall sensor?

15) Figure 5: I think the name of signals should be included in the caption.

Author Response

Dear Reviewer,

Thank you for spending precious time on our previous manuscript and your valuable suggestions. The responses are detailed following each comment in the attached document. Necessary explanations have been added in the revised paper.

Best regards

Maciej Sulowicz

Reviewer 2 Report

Practical Adaptation of a Low-Cost Voltage Transducer with an Open Feedback Loop for Precise Measurement of Distorted Voltages

Reviewer’s comments:

1.      As the paper title shows, this is a “Practical Adaptation of a Low-Cost Voltage Transducer with an Open Feedback Loop”. However, there is no discussion about cost. Readers will be confused.

2.      The research survey is not enough. Most of the articles listed in References are old. Of course, Refs.[9-11] are new articles. However, as you can see from Sect.1, the relation between the proposed technique and these articles is very small. You must survey past studies in detail. Besides, you must justify the effectiveness of the proposed method by comparing with other latest methods.

3.      Don’t use acronym, such as DAQ (Data AcQuisition), FEMM (Finite Element Method Magnetics), etc., without explanation.

4.      The problem definition of this work is not clear. In the introduction part, the drawbacks of each conventional technique should be described clearly. You should emphasize the difference with other methods to clarify the position of this work further.

5.      The explanation about the mathematical formulas is not enough. Furthermore, the meaning of parameters is not clear. Readers will be confused.

6.      You should unify the mathematical expression. e.g. In Eq. (1), the magnetic permeability is defined as μ. However, the magnetic permeability is defined as μ in p.4, l. 138.

7.      In p.5, l.160, “after the integration (5), …” must be “after the integration (6), …”.

8.      How did you get Eq. (8) for Eq. (7)? Eq.(8) seems a wrong equation.

9.      In Sect.4, the experimental conditions are nor clear. In Fig.1 (b), you should describe the value of resistor, capacitor, etc.

10.   The effectiveness of the proposed technique is not clear. In Sect.4, compare the proposed technique with existing techniques. In this paper, the comparison was performed between the proposed technique and Ref. [24] only. Ref. [24] is not an article but an application note. You should emphasize the difference with existing techniques. Add more comparison data.

Author Response

(The authors gave the same response as above.)

Reviewer 3 Report

Observations:

-Page 2, Rows 75-76: Why are you choosing A 3515 LUA Hall Sensor?

-Page 3, Row 94: Why considered N = 1500 turns?

-Pag.4, Row 143: Where is Bd in Eq. (1)? Put reference for Eq.(1);

-Pag.5 Fig.2: Put more lines into the grid (vertical and horizontal axes). It is only one.

-Pag.7, Row 234: Why are you choosing those values for Rd1 and Rd2?

- Pag.8, Fig.5: How do you explain phase shift with increasing frequency?

-Pag.9, Row 272: k=39?

- Pag.13, Table 3: Why are you choosing for experiments Rd 22k ?

- Which are the main disadvantages of the proposed sensor copared with other Hall sensors?

Author Response

(The authors gave the same response as above.)

Round 2

Reviewer 2 Report

Practical Adaptation of a Low-Cost Voltage Transducer with an Open Feedback Loop for Precise Measurement of Distorted Voltages

In this paper, the authors presented the project proposal of a low-cost transducer with a Hall-effect sensor placed in a ferromagnetic core’s air gap. It enables the measurement of the distorted voltage instantaneous values without the feedback loop used for measurements in electrical machines. The revised version is well written and organized paper, I think. It is scientifically sound and contains sufficient interest to merit publication.